# Brainstem BDNF neurons are downstream of GFRAL/GLP1R signalling

Claire H. Feetham[1,3], Valeria Collabolletta[1,3], Amy A. Worth [1], Rosemary Shoop[1], Sam Groom[1], Court Harding[1], Mehdi Boutagouga Boudjadja[1], Tamer Coskun[2], Paul J. Emmerson[2], Giuseppe D'Agostino [1] & Simon M. Luckman [1] ✉

Growth differentiation factor 15, GDF15, and glucagon-like peptide-1 (GLP-1) analogues act through brainstem neurons that co-localise their receptors, GDNF-family receptor α-like (GFRAL) and GLP1R, to reduce food intake and body weight. However, their use as clinical treatments is partially hampered since both can also induce sickness-like behaviours, including aversion, that are mediated through a well-characterised pathway via the exterolateral parabrachial nucleus. Here, in mice, we describe a separate pathway downstream of GFRAL/GLP1R neurons that involves a distinct population of brain-derived neurotrophic factor (BDNF) cells in the medial nucleus of the tractus solitarius. Thus, BDNF[mNTS] neurons are required for the weight-reducing actions of both GDF15 and the GLP1RA, Exendin-4. Moreover, acute activation of BDNF[mNTS] neurons is sufficient to reduce food intake and drive fatty acid oxidation and might provide a route for longer-term weight loss.

The cytokine, growth differentiation factor 15 (GDF15), is expressed and secreted by tissues when they are stressed or damaged[1,2]. It acts through its only identified receptor, GDNF-family receptor α-like (GFRAL) to reduce food intake and body weight[3–6]. Much evidence points towards GDF15 not being involved in homeostatic regulation of whole-body metabolism: GDF15 and GFRAL knock-out mice, or mice treated with GFRAL-blocking antibodies, generally have normal food intake and body weight[7–10]. Instead, GDF15/GFRAL signalling plays an adaptive role in response to aversive stimuli, tissue damage and/or disease, causing protective, sickness-related changes in behaviour and metabolism[11–13] (reviewed by ref. 14). Thus, administration of exogenous GDF15 produces anorexia, aversion, conditioned taste avoidance (CTA) and pica in various animal models[10,11,15]. In addition, GDF15 causes triglyceride release from adipose tissue or liver and increases fatty acid oxidation in the heart and other striated muscle as part of an allostatic response[6,13,16]. In fact, weight loss due to prolonged treatment with exogenous GDF15 is dependent on a maintenance in energy expenditure and fatty acid oxidation in the face of reduced body mass[16].

The GDF15 receptor, GFRAL, is located exclusively in the brainstem[3–6]. GFRAL neurons are primarily glutamatergic/

cholecystokininergic and form a continuum between the area postrema (AP) and the dorsal nucleus of the tractus solitarius (NTS)[10]. They are also a sub-population of the neurons in this region that contain the receptor for glucagon-like peptide 1 (GLP-1)[17–19]. Although the anorectic effects of GDF15 and GLP-1 are independent of each other's receptors[6,17,20], genetic silencing of CCK[AP/NTS] neurons blocks the anorectic effects of both GDF15 and clinically important GLP-1 receptor agonists (GLP1RA)[10,19]. Henceforth, these GFRAL/CCK/GLP1R-expressing glutamatergic cells in the AP/NTS will be termed GFRAL neurons.

In addition to activating GFRAL neurons, exogenous GDF15 or GLP1RA also activate downstream, non-GFRAL neurons that are part of a well-characterised aversive pathway, which includes calcitonin gene-related peptide (CGRP) neurons in the exterolateral parabrachial nucleus (elPBN) and protein kinase C delta type (PKCδ) neurons in the central amygdala (CeA) and oval bed nucleus of the stria terminalis (BSTov)[10,21–23]. GFRAL neurons activated by GDF15 project directly to the elPBN, but not to forebrain structures[10], thus this direct pathway is presumed responsible for the aversive effects of GDF15/GLP1R signalling[10,19,24]. While GFRAL neurons mediate the aversive response to GLP1RA[19,25], it is less clear whether they are also responsible for all weight-managing effects of those agonists since there are other GLP1R-

[1]Faculty of Biology, Medicine and Health, University of Manchester, Manchester, UK. [2]Lilly Research Laboratories, Eli Lilly & Company, Indianapolis, USA. [3]These authors contributed equally: Claire H. Feetham, Valeria Collabolletta. ✉e-mail: simon.luckman@manchester.ac.uk

containing cells in this part of the brainstem[18,26,27], as well as other regions of the brain[28]. The changes in nutrient utilisation induced by GDF15 appear also to involve engagement of the sympathetic nervous system[12,13,16] and/or possibly adrenal glucocorticoids[29]. Importantly, corticotrophin-releasing hormone (CRH) neurons, which lie at the head of the sympathetic and hypothalamo-adrenal axes, are activated by both GDF15[10] and GLP1RA[30]. However, the pathways which lead to sympatho-adrenal activation and any correlated weight loss are unclear.

Here, we utilise a *Gfral*[Cre] mouse model which faithfully targets GFRAL neurons in the AP/NTS. GFRAL neurons send dense fibre projections within the dorsal vagal complex and to the elPBN, but not to other parts of the brain. The chemogenetic activation of GFRAL neurons with a designer receptor causes obvious sickness-like behaviour: an inhibition of food intake and gastric emptying, a conditioned aversion[24], as well as a transient reduction in energy expenditure and body temperature, and a sustained reduction in respiratory exchange ratio (RER), a measure of increased fatty acid oxidation[31]. While showing optogenetic activation of the GFRAL[AP/NTS]→elPBN pathway is sufficient to induce anorexia, we also identify a population of brain-derived neurotrophic factor neurons in the medial NTS (BDNF[mNTS]) downstream of GFRAL cells, which are required for the weight-reducing actions of both GDF15 and the GLP1RA, Exendin-4 (EX4). Interestingly, selective activation of BDNF[mNTS] neurons also causes profound effects on food intake and RER. Thus, BDNF[mNTS] neurons may have a wide function in whole-body physiology and, more specifically, may have an important role in mediating the weight-reducing effects of clinically relevant GLP1RA.

## Results

### GFRAL neuron connections are confined to the brainstem

Using homologous recombination in embryonic stem cells, we generated a transgenic mouse expressing IRES-Cre recombinase under the *Gfral* gene promoter. To confirm faithful, cell-specific expression, and to visualise cell projections, the *Gfral*[Cre] mouse was crossed with a Cre-dependent channel rhodopsin-eYFP mouse (Fig. 1a). Adult mice were culled and dual-label immunohistochemistry carried out for enhanced yellow fluorescent protein (eYFP) and native GFRAL. This revealed an excellent co-localisation in the AP and NTS (Fig. 1b), with no significant expression elsewhere in the brain. On average, there were $28 \pm 3$ *Gfral*[Cre:ChR2-eYFP] cells per section in the AP and $37 \pm 2$ in the dorsal NTS (at the same rostro-caudal extent as the AP). There was a very small number of eYFP-positive (+ve) cell bodies in the laterodorsal tegmental nucleus (LDTg) that resemble neurons, and a few cells in the pyramidal tracts/olivary complex which resemble oligodendrocytes (Supplementary Fig. 1a). There were less than three cells in any section at either of these sites and the cells do not contain native GFRAL staining, nor did they respond to administration of exogenous GDF15.

Exogenous GDF15 injected subcutaneously produced a strong activation of *Gfral*[Cre:ChR2-eYFP] cells in the AP (saline $8 \pm 4\%$, GDF15 $60 \pm 5\%$; Students unpaired *t*-test $p < 0.0001$) and NTS (saline $4 \pm 2\%$, GDF15 $32 \pm 3\%$; $p < 0.0001$; Fig. 1c). In the GDF15-treated mice, $58 \pm 6\%$ of FOS+ve cells in the AP contained eYFP. However, in the NTS this figure was only $7 \pm 1\%$. Previously, we noted that administration of GDF15 induces FOS in a group of unidentified, GFRAL-negative (-ve) neurons in the mNTS, some of which project directly to the paraventricular nucleus of the hypothalamus (PVH)[10], which we presume are directly downstream of the nearby GFRAL neurons. Indeed, eYFP fibres were clearly visible in the mNTS in close proximity with FOS+ve/eYFP-ve neurons (Fig. 1d). As the majority of GFRAL[AP/NTS] neurons contain vesicular glutamate transporter and CCK, they are assumed to make excitatory connections with post-synaptic target neurons[10,18]. We made injections of the retrograde tracer FluoroGold (FG) into the mNTS of wild-type mice and subsequently injected them with vehicle

or GDF15. There were FG-containing, GDF15-activated cell bodies visible in the AP (Supplementary Fig. 1b), providing evidence for the local connection, with the caveat some of these AP cells may possibly have been accidentally contaminated by the mNTS FG injections. There also is a possibility that some mNTS neurons can respond to descending input from higher-order cells in the elPBN, PVH, CeA or BSTov, which we have previously shown to be activated by exogenous GDF15[10]. There was FOS and FG within a small number of mNTS-projecting PVH cells (in both vehicle- and GDF15-injected mice), but none in the PBN, CeA or BSTov, suggesting that these regions are not driving the activity in the mNTS (Supplementary Fig. 1b).

### An anorectic pathway between GFRAL neurons and the PBN

In our original study, we demonstrated that GFRAL neurons, that are activated by exogenous GDF15, project directly to the elPBN, where they activate CGRP (gene name *Calca*) neurons; but they do not project directly to other forebrain sites[10]. Furthermore, *Gfral*[Cre:ChR2-eYFP] fibres are close to GDF15-induced FOS in the elPBN but not in other regions of the parabrachial nucleus (Fig. 1d). Ref. 24 found that disabling *Calca*[Cre] neurons with tetanus toxin attenuated the anorectic response, but completely blocked the aversion to exogenous GDF15. CGRP[elPBN] neurons are part of a well-characterised interconnected pathway linking the brainstem to the forebrain, including the CeA[21-23]. *Calca*[Cre-GFP] mice were targeted with FG into the CeA and subsequently given GDF15. Approximately, 20–30% of CGRP[elPBN] neurons projected to the CeA, and about 15% of FG-stained neurons contained *Calca* (Supplementary Fig. 2a). Mice were also injected with FG into the elPBN and subsequently with GDF15. As expected[10], GDF15-activated neurons project from the AP/NTS directly to the elPBN. Although there were also many cells in the PVH, CeA and BSTov which descend to the elPBN, only a small number of these contained GDF15-induced FOS (Supplementary Fig. 2c).

To confirm that this is a functional connection between GFRAL neurons and the PBN, GFRAL terminals were activated using optogenetic stimulation. *Gfral*[Cre] mice were crossed, as before, with a Cre-dependent channel rhodopsin-eYFP mouse, and both wild-type (*Gfral*[WT:ChR2-eYFP]) and Cre+ve (*Gfral*[Cre:ChR2-eYFP]) littermates had a single optic fibre implanted above the elPBN on one side (Fig. 2a). Following recovery from surgery, mice were fasted overnight and, in the morning, were tethered to allow optical stimulation of nerve fibres in the PBN. Subsequent immunohistochemical validation showed that this induced FOS expression in the cells of the elPBN (Fig. 2b) but not in GFRAL cell bodies (Supplementary Fig. 2d). Food intake was measured during 10 Hz stimulation. There was a significant reduction in fast-induced food intake in the *Gfral*[Cre:ChR2-eYFP] mice when compared with wild-type *Gfral*[WT:ChR2-eYFP] littermates (Fig. 2c), indicating a direct anorectic pathway from GFRAL neurons to the PBN. For comparison, there was no difference in food intake between *Gfral*[Cre:ChR2-eYFP] and *Gfral*[WT:ChR2-eYFP] littermates if the mice were tethered but no photostimulation applied (Supplementary Fig. 2e).

### Activation of GFRAL neurons produces sickness-like behaviour

Administration of exogenous GDF15 causes anorexia and CTA[10,11,15]. However, it is not clear if the two can be dissociated. Likewise, high concentrations of exogenous GDF15 reduce gastric emptying which will indirectly affect food intake[32,33]. We crossed our *Gfral*[Cre] and a stimulatory designer receptor mouse (Fig. 3a)[34] and treated the adult offspring with either saline or the designer drug, clozapine N-oxide (CNO), in a crossover design. As well as inducing FOS in GFRAL[AP/NTS] neurons and in non-GFRAL neurons in the NTS (Fig. 3b), activation of *Gfral*[Cre:hM3Dq-mCherry] neurons caused the strong induction of FOS in non-GFRAL neurons in the PBN, CeA, BSTov, PVH, supraoptic nucleus (SON), parasubthalamic nucleus (PSTN) and infralimbic cortex (IC) (Supplementary Fig. 3a). This pattern closely resembles that following administration of exogenous GDF15[10].

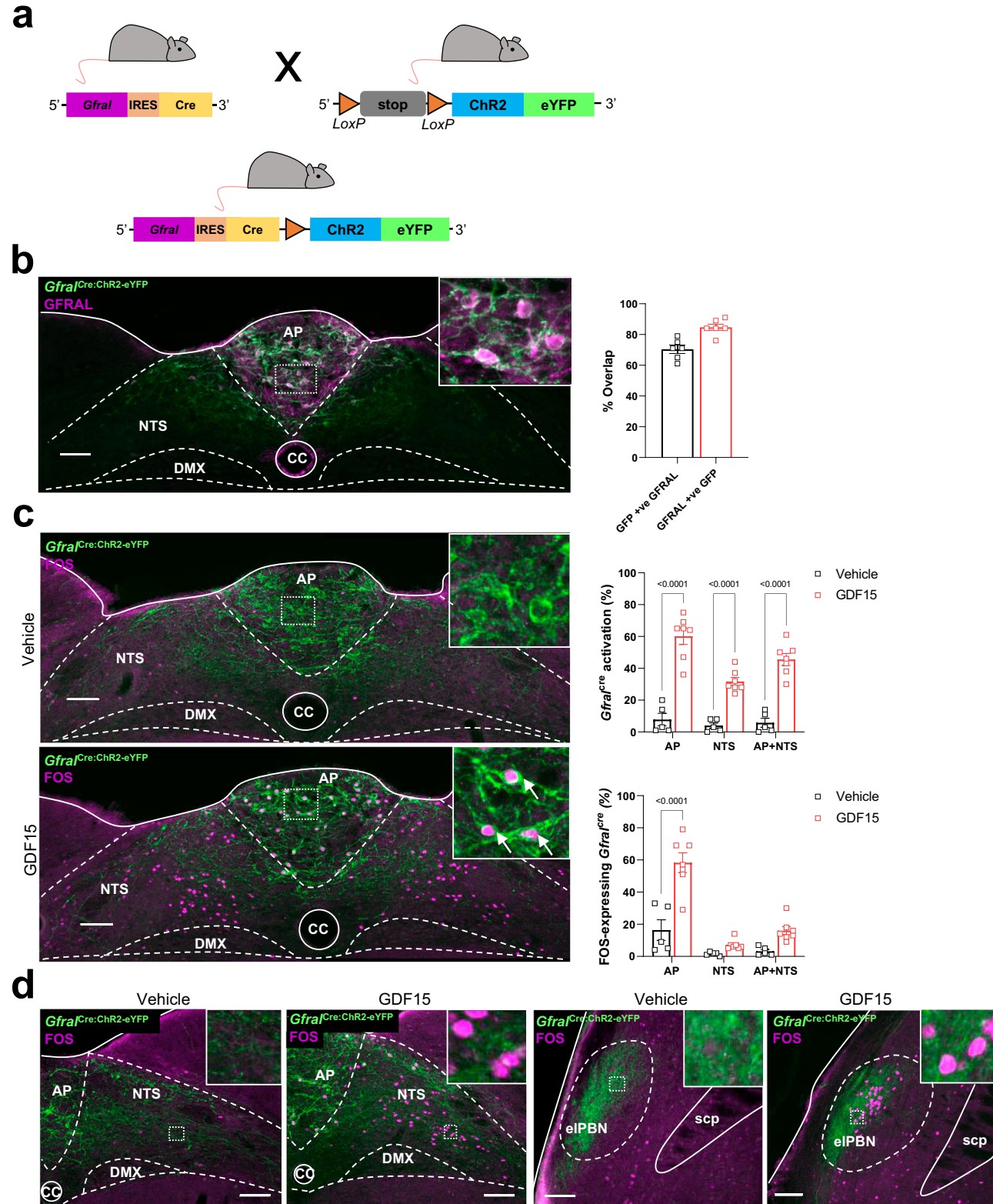

Chemogenetic activation showed a very robust inhibition of normal, night-time food intake which lasted for several hours, as well as a significant reduction in body weight and gastric emptying (Fig. 3c, d). Following CNO injection, mice showed obvious signs of sickness behaviour (i.e. hunched, immobile), and using a CTA test (Fig. 3e), we confirmed that activation of *Gfral* neurons causes avoidance[24,25,31]. To some extent, these results are as predicted, as we and others have already reported that GFRAL expression overlaps with CCK[10] and that

artificial activation of CCK[AP/NTS] neurons causes both anorexia and CTA[35,36]. Importantly, as reported recently[31], selective activation of GFRAL neurons during the day had a transient effect on thermogenesis (a reduction in energy expenditure and body temperature, measured by indirect calorimetry and thermal imaging, respectively), and a more sustained effect on nutrient utilisation (reduced RER; Fig. 3f–h). These changes in metabolism are expected as the animals have reduced activity and, as they have stopped eating, they convert towards

**Fig. 1 | GFRAL neurons and their connections are confined to the brainstem.**
**a** Schematic illustrating the cross between a *Gfral*[Cre] and a Cre-dependent channel rhodopsin-eYFP mouse. **b** Dual-fluorescence labelling for GFRAL (magenta) with eYFP (green) in *Gfral*[Cre:ChR2-eYFP] mice. Note that the GFRAL antibody has been validated as staining is not present in null *Gfral*[-/-] mice. The percentage co-localisation GFP+ve/GFRAL+ve neurons (black squares) and GFRAL+ve/GFP+ve neurons (red squares) in the AP and NTS of *Gfral*[Cre:ChR2-eYFP] mice is presented on the right (male and female 12-weeks old; *n* = 6). **c** Dual-fluorescence labelling showing FOS expression (magenta) with eYFP (green) in the AP and NTS of *Gfral*[Cre:ChR2-eYFP] mice following GDF15 (4 nmol kg⁻¹, SC). White arrows in higher magnification insets indicate co-labelled cells. The percentage of activated *Gfral*[Cre:ChR2-eYFP] neurons and of FOS+ve/GFRAL+ve neurons is presented on the right (vehicle black squares, *n* = 5; red squares, *n* = 7; ****p < 0.0001, two-way ANOVA followed by Sidak's multiple comparisons test). **d** Dual-fluorescence labelling showing GDF15-induced FOS expression (magenta) in a group of non-identified GFRAL-ve neurons, in the medial NTS or elPBN of *Gfral*[Cre:ChR2-eYFP] mice (eYFP in green; vehicle *n* = 5, GDF15 *n* = 7). AP (area postrema), CC (central canal), DMX (dorsal motor nucleus of the tenth cranial nerve, vagus), elPBN (exterolateral parabrachial nucleus), NTS (nucleus of the tractus solitarius), scp (superior cerebellar peduncle). All data presented as mean ± SEM. Scale bars indicate 100 μm. Source data are provided as a Source Data file.

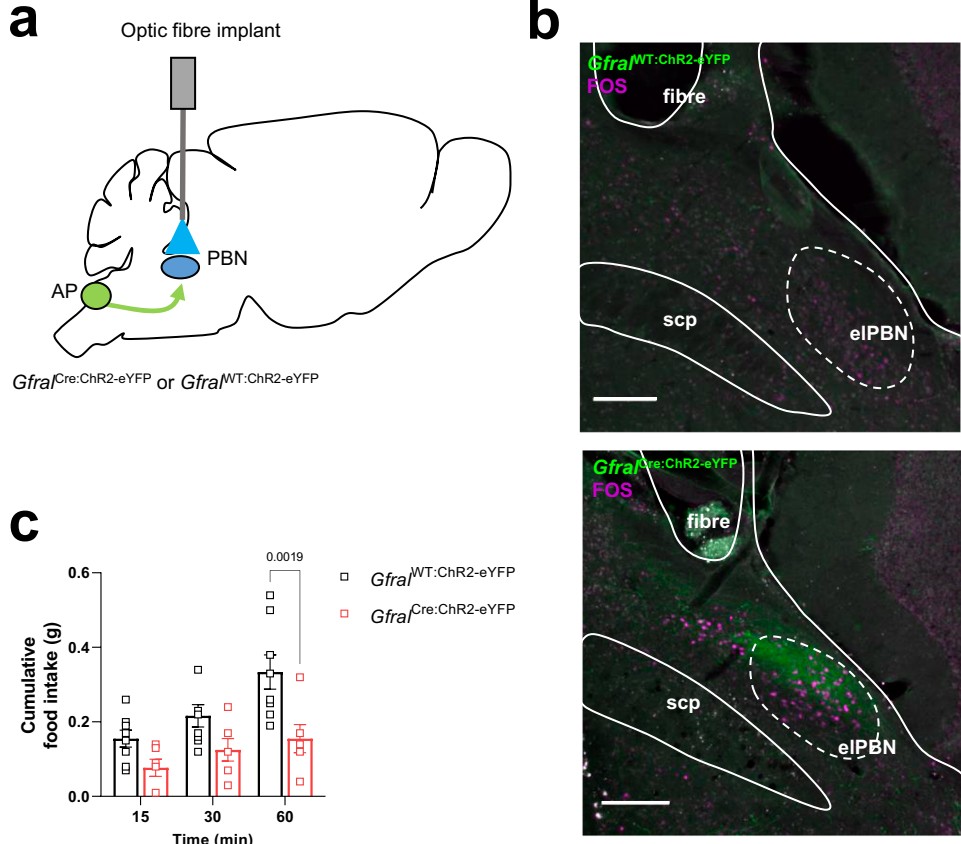

**Fig. 2 | An anorectic pathway between GFRAL neurons and the parabrachial nucleus. a** Schematic illustrating fibre optic implant above the elPBN of *Gfral*[Cre:ChR2-eYFP] mice and wild-type littermates, *Gfral*[WT:ChR2-eYFP]. Based on the Paxinos and Franklin Mouse Brain Atlas[56]. **b** Dual-fluorescence labelling showing FOS expression (magenta) close to green (eYFP) fibres of *Gfral*[Cre:ChR2-eYFP] mice (bottom) and *Gfral*[WT:ChR2-eYFP] (top) following 10 Hz optogenetic stimulation of GFRAL terminals in the elPBN. **c** Cumulative food intake in overnight fasted *Gfral*[Cre:ChR2-eYFP] (red squares; *n* = 6) and *Gfral*[WT:ChR2-eYFP] (black squares; *n* = 8) mice during 10 Hz optogenetic stimulation for 60 min in the elPBN (**p = 0.0019; two-way ANOVA followed by Sidak's multiple comparisons test). AP (area postrema), elPBN (exterolateral parabrachial nucleus), scp (superior cerebellar peduncle). All data presented as mean ± SEM. Scale bars indicate 100 μm. Source data are provided as a Source Data file.

metabolising fat rather than carbohydrate (Fig. 3g). Indeed, the derived value for fat oxidation increased significantly over a 12-h period (Supplementary Fig. 3b). Interestingly, a change to fat utilisation has also been reported in response to repeat dosing with exogenous GDF15, including in pair-fed animals[6,16], and has been suggested as an important response to GDF15 in models of both septicaemia and cancer[12,13]. Of note, since repeated administration of GDF15 leads to sustained fatty acid oxidation without long-term anorexia[16], GFRAL neurons may be able to regulate the two independently. With this in mind, we determined whether we could dissociate the effect on RER from any negative energy balance caused by reduced food intake by repeating acute GFRAL neuron activation during the daytime in mice that had no food available. *Gfral*[Cre] mice crossed with the stimulatory designer receptor

mice were prepared as before. On the morning of experimentation, food was removed from the cage and an hour later the mice were injected with saline or CNO in a crossover design. As expected, RER fell immediately when food was removed, suggesting this is not due to a sudden negative energy balance, but is probably neurally mediated. After saline injection, the RER plateaued, whereas after CNO, the RER continued to reduce significantly and stayed low until food was returned (Supplementary Fig. 3c). Together, this result and data from long-term GDF15 treatment[16] suggest that an increase in fatty acid oxidation induced by the activation of GFRAL neurons is not completely secondary to reduced energy intake. However, a caveat remains that a reduction in gastric emptying may also be impinging on the entry of nutrients to the bloodstream, exacerbating the RER response. It is

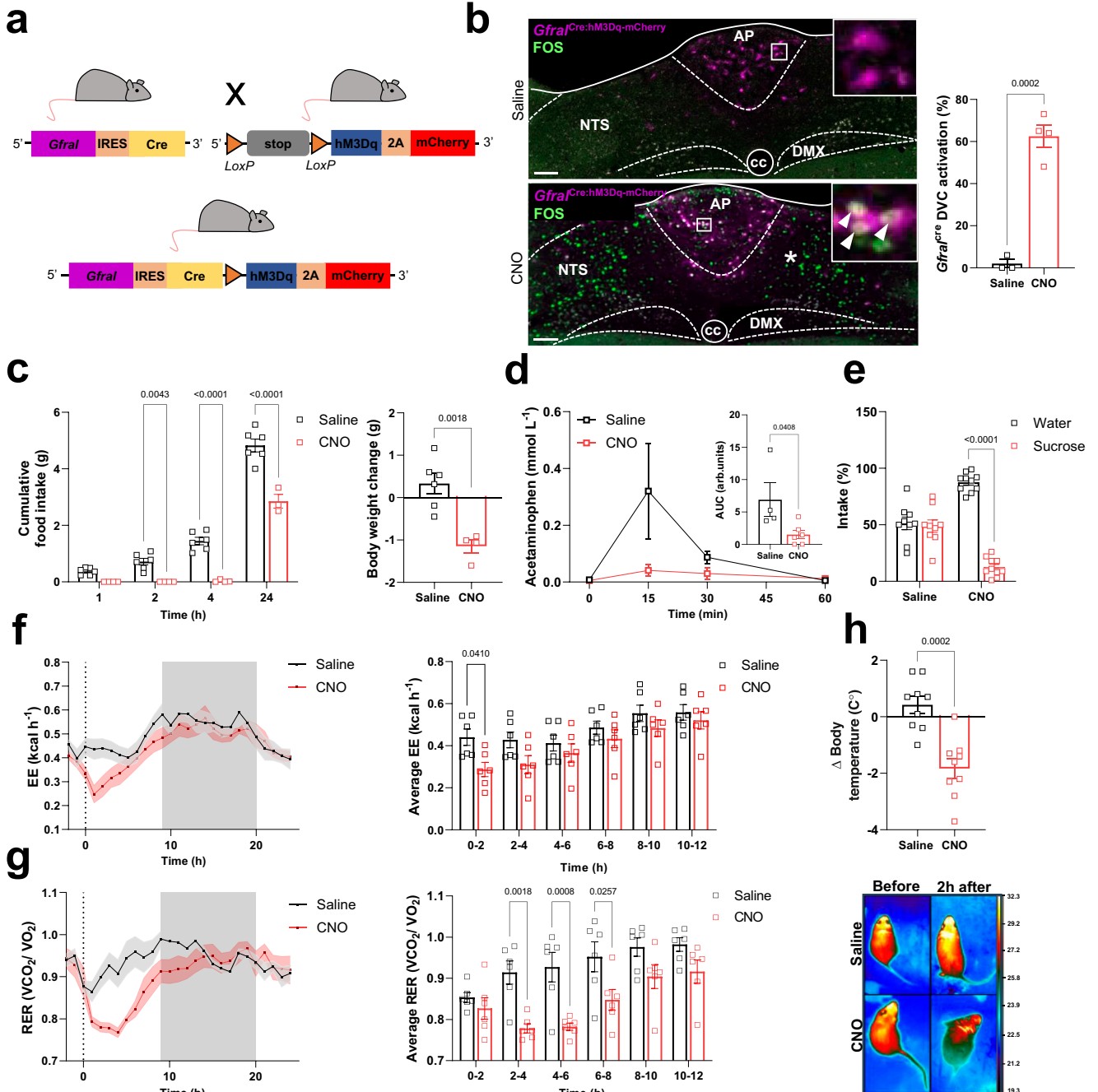

**Fig. 3 | Faithful activation of GFRAL neurons produces sickness-like behaviour.**
**a** Schematic illustrating the *Gfral*Cre cross with the loxSTOPlox-hM3Dq-mCherry reporter mouse. **b** Chemogenetic stimulation of *Gfral*Cre:hM3Dq-mCherry offspring with CNO (3 mg kg[-1]) induces FOS (green) in GFRALAP/NTS neurons (magenta) and also in unidentified neurons that lie downstream in the mNTS (asterisk). The percentage of activated GFRAL neurons in the dorsal vagal complex (DVC) is presented on the right (saline *n* = 3; CNO *n* = 4; ***p* = 0.0002; two-tailed Student's unpaired *t*-test). **c** CNO treatment of *Gfral*Cre:hM3Dq-mCherry mice decreases cumulative night-time food intake and body weight (saline *n* = 6; CNO *n* = 4; ***p* = 0.0043, ***p* = 0.0018, *****p* < 0.0001; two-way ANOVA followed by Sidak's multiple comparisons test or two-tailed Student's unpaired *t*-test). **d** Gastric emptying, as measured by the appearance of acetaminophen in the blood, is also reduced. Area under the curve (AUC) for plasma acetaminophen from 0 to 60 min after oral gavage (saline *n* = 4, CNO *n* = 6; **p* = 0.0408; two-tailed Student's unpaired *t*-test). **e** Activation of *Gfral*Cre:hM3Dq-mCherry neurons causes aversion, as measured by a conditioned taste

avoidance test (saline *n* = 9, CNO *n* = 10; *****p* < 0.0001; two-way ANOVA followed by Sidak's multiple comparisons test). **f** CNO treatment of *Gfral*Cre:hM3Dq-mCherry mice induced a transient decrease in energy expenditure that lasted for about 2 h (*n* = 6 per group; **p* = 0.0410), and (**g**) a decrease in RER that lasted for about 10 h (**p* = 0.0257, ***p* = 0.0018, ****p* = 0.0008; two-way ANOVA followed by Sidak's multiple comparisons test). Indirect calorimetry was used to measure metabolic gases. **h** Top, body temperature measurement by thermal imaging in *Gfral*Cre:hM3Dq-mCherry mice before and 2 h after saline or CNO (*n* = 9 per group ***p* = 0.0002; two-tailed Student's unpaired *t*-test). Thermal images of two *Gfral*Cre:hM3Dq-mCherry mice before and after saline or CNO injection. arb.units (arbitrary units), AP (area postrema), CC (central canal), DMX (dorsal motor nucleus of the tenth cranial nerve, vagus), NTS (nucleus of the tractus solitarius). All data presented as mean ± SEM. Saline black squares and CNO as red squares throughout. Source data are provided as a Source Data file.

impossible to totally dissociate the effect on RER from reduced nutrient availability, but it remains interesting that a drop in RER following either removal of food or GDF15 may activate similar neural mechanisms.

## NTS cells downstream of GFRAL contain BDNF

We wished to further investigate the non-GFRAL cells in the mNTS activated by GDF15, which we previously found do not contain transmitters that are commonly associated with brainstem regulation of food intake: catecholamines (tyrosine hydroxylase), CCK, prolactin-releasing peptide (PrRP), GLP-1 or proopiomelanocortin[10]. In a survey of NTS neurons, Garfield and colleagues[37] noted a population of brain-derived neurotrophic factor (BDNF) cells which occupy the position of the group of GDF15-activated neurons we describe. These BDNF[mNTS] neurons do not contain the leptin receptor[37] and, therefore, are distinct from a sub-set of leptin-receptor neurons which input onto GFRAL neurons[38]. BDNF signalling within the dorsal vagal complex can affect feeding[39–41]. Thus, we tested if BDNF[mNTS] neurons are relevant to our studies by carrying out triple-label RNAscope. *Bdnf* mRNA was expressed along with *Gfral* in the AP, in the population of *Gfral*+ve neurons that exists just outside the AP in the dorsal NTS[10], but also in *Gfral*-ve cells in the mNTS in the region where we see FOS induction after stimulation of GFRAL neurons. As well as activating *Gfral*+ve neurons, GDF15 caused c-*fos* induction in *Gfral*-ve/*Bdnf*+ve neurons in the mNTS (Fig. 4a, b). Interestingly, the BDNF[mNTS] neurons were also activated by the GLP1RA, EX4 (Fig. 4c), supporting a common relay for certain blood-borne anorectic signals. $41 \pm 2\%$ and $31 \pm 2\%$ of c-*fos*-expressing cells in the NTS following GDF15 or EX4, respectively, were *Bdnf* +ve neurons. Next, we demonstrated EX4 acts on BDNF neurons in the AP and NTS using ex vivo, GCaMP-dependent calcium imaging. Since the primary sensitive neurons in the AP contain GFRAL, GLP1R and BDNF[18], both *Gfral*[Cre:GCaMP6] and *Bdnf*[Cre:GCaMP6] cells in the AP responded strongly to EX4, as expected with a robust increase in intracellular calcium (Fig. 4d, e). By contrast, *Bdnf*[Cre:GCaMP6] cells in the mNTS did not respond to application of EX4 with a large increase in intracellular calcium (Fig. 4f) suggesting that, though they can be activated synaptically, BDNF[mNTS] neurons are not a direct target for GLP1RA. RNASeq has shown that there are five *Bdnf*-containing neuronal clusters in the dorsal vagal complex, one which corresponds with the GFRAL[AP/NTS] cells, and another that corresponds with PrRP/calcitonin receptor (*Calcr*) neurons in the NTS[27]. We have previously published PrRP neurons are not activated by GDF15[10] and, in a separate experiment, we now show that PrRP neurons are not activated by EX4 either (Supplementary Fig. 4). Thus, we have identified a distinct BDNF[mNTS] population activated downstream of GFRAL cells.

## BDNF[mNTS] neurons are required for the metabolic effects of GDF15 or EX4

There are clear similarities between the actions of GDF15 and GLP1RA, to reduce food intake, RER and body weight[6,16,42–45]. Therefore, we measured these parameters in mice which had their BDNF[mNTS] neurons selectively disabled. *Bdnf*[Cre] mice were injected bilaterally into the mNTS with adeno-associated virus (AAV) encoding the tetanus toxin light chain (*Bdnf*[Cre:NTS-TeNT]) or an AAV-mCherry control (*Bdnf*[Cre:NTS-mCherry]) (Fig. 5a; Supplementary Fig. 5a). Tetanus toxin blocks transmitter release from transduced neurons effectively disabling them from acting on their downstream targets. Disabling BDNF[mNTS] neurons with tetanus toxin caused an increase in body weight gain in male mice, but this did not reach significance in females (Supplementary Fig. 5b, c). The increased body weight gain in the male mice was due to increased daily food intake and feed efficiency (weight gain per food intake), rather than a change in energy expenditure (Supplementary Fig. 5d–f), suggesting these neurons may play a role in homeostatic metabolic regulation.

We first treated our model with GDF15 at lights out. *Bdnf*[Cre:NTS-mCherry] control mice had reduced body weight measured at 24 h after

injection of GDF15, that was not observed in tetanus toxin *Bdnf*[Cre:NTS-TeNT] mice (Fig. 5b). This correlated with attenuated effect on food intake and a loss of any significant effect on RER in *Bdnf*[Cre:NTS-TeNT] mice (Fig. 5c, d). Similarly, the effect of EX4 on body weight and food intake was reduced in separate cohorts of *Bdnf*[Cre:NTS-TeNT] mice treated at night (Fig. 5e, f and results presented in the next section, Fig. 6b, c). Finally, we also treated our model with EX4 in the daytime when mice are normally eating little. Again, the effects on RER were reduced in *Bdnf*[Cre:NTS-TeNT] mice (Fig. 5g). Interestingly, the attenuation of the effect on RER was much more marked than the attenuation of the anorexia or energy expenditure (Supplementary Fig 5g, h). Since we and others have published previously that the GFRAL to elPBN pathway is required for the aversive response to clinically relevant GLP1RA[18,19], we asked whether this was true also for the mNTS pathway. Both *Bdnf*[Cre:NTS-mCherry] control mice and *Bdnf*[Cre:NTS-TeNT] mice responded to GLP1R agonism by forming a CTA (Supplementary Fig. 6), suggesting that this alternative pathway is not required for the nausea effects of GLP1RA. Our results fit with a model whereby the effects of GDF15 and GLP1RA act partly through GFRAL neurons projecting to BDNF[mNTS] neurons, and partly other projections, including to CGRP[PBN] neurons.

## Disabling BDNF[mNTS] neurons prevents EX4-mediated activation of the PVH, but not other downstream targets

As noted, systemic injection of GDF15 or GLP1RA induces very similar patterns of brain activation, including in the elPBN and PVH[10,30], which we propose form distinct pathways downstream of GFRAL neurons. To test this further, we carried out a separate night-time experiment using *Bdnf*[WT] control and *Bdnf*[Cre] littermates transfected with tetanus toxin (Fig. 6a; Supplementary Fig. 7a). Mice were injected at lights out and left undisturbed for 24 h. *Bdnf*[WT] controls, but not *Bdnf*[Cre:NTS-TeNT] mice, responded to EX4 with a significant reduction in body weight and food intake (Fig. 6b, c). Following a period of recovery, the same two groups of mice were split and given injections of either saline or EX4 before perfusion and processing of brain sections. As we predicted, injection of EX4 into *Bdnf*[WT] control mice induced FOS in the AP and mNTS, as well as in the elPBN, PVH, CeA, BSTov, PSTN and IC (Fig. 6d, e; Supplementary Fig. 7b–e). Likewise, FOS was induced in the same brain regions of *Bdnf*[Cre:NTS-TeNT] mice following EX4, including in the elPBN (Fig. 6d; Supplementary Fig. 7b–e). The only exception was in the PVH, where the induction of FOS by EX4 was severely attenuated in *Bdnf*[Cre:NTS-TeNT] mice (Fig. 6e). These data support our previous finding that GFRAL neurons project directly to the elPBN but indirectly to the PVH[10]. Thus, we have identified an independent pathway which mediates some of the effects of both GDF15 and GLP1RA.

## Acute activation of BDNF[mNTS] neurons is sufficient to increase fatty acid oxidation

To determine if they are sufficient to reduce food intake and RER, we transduced BDNF[mNTS] neurons bilaterally with a Cre-dependent AAV-hSyn-DIO-hM3Dq-mCherry (Fig. 7a, b). Stimulation of *Bdnf*[Cre:NTS-hM3Dq] neurons chemogenetically with a low dose of CNO (0.03 mg/kg) significantly reduced body weight, food intake and RER (Fig. 7c, f, g). Interestingly, although RER was reduced robustly, this stimulus produced no overall effect on either energy expenditure or body temperature (Fig. 7d, e). Thus, although there was a reduction in body weight, these results indicate that fatty acid oxidation may possibly occur independently of changes in energy expenditure and/or thermogenesis. However, stronger activation of BDNF[mNTS] neurons produced progressively greater effects on RER, as well as reductions in energy expenditure and body temperature, as if the mice were progressing towards a state of torpor (Fig. 7c–e).

## Discussion

The actions of GDF15 are adaptive in the face of tissue damage or infection/illness[14]. This includes sickness and aversive behaviours which

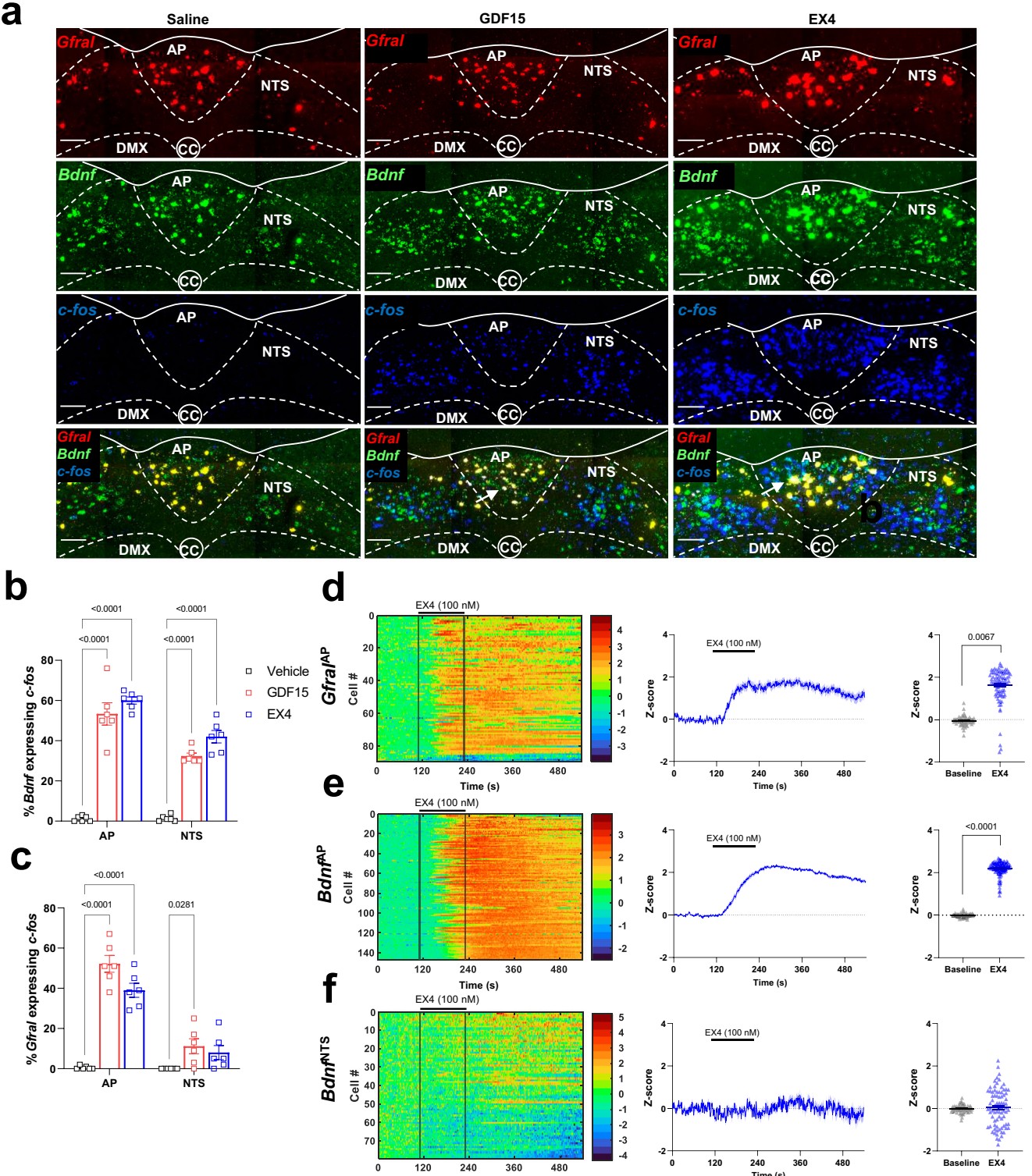

**Fig. 4 | NTS cells downstream of GFRAL contain BDNF. a** Triple-label RNAscope analysis showing *Bdnf* (green), *Gfral* (red) and c-*fos* (blue) mRNAs co-localise in the AP and NTS following treatment with either GDF15 or EX4. White arrows indicate triple-labelled cells. Within the NTS there is also a number of activated *Bdnf*+ve/*Gfral*-ve neurons. **b, c** Percentage of *Gfral* or *Bdnf* neurons activated by GDF15 and EX4 in the AP and NTS (vehicle black squares, GDF15 red squares, EX4 blue squares; $n = 6$ per group; *$p = 0.0281$, ****$p < 0.0001$; two-way ANOVA with Dunnett's multiple comparisons test; presented as mean ± SEM). **d–f** Ex vivo measurement of intracellular calcium in brainstem slices following treatment with 100 nM EX4. Left,

heat maps showing responses (expressed as Z-score) of all *Gfral*AP (89 cells, 4 slices, 3 animals), *Bdnf*AP (147 cells, 4 slices, 3 animals) and *Bdnf*NTS (79 cells, 4 slices, 2 animals) neurons recorded; middle, averaged calcium responses from all cells over time (mean ± 95% CI); right, peak calcium response before and after EX4 application averaged per slice (Baseline grey triangles, EX4 blue triangles, $n = 4$ brain slices per group; **$p = 0.0067$, ****$p < 0.0001$; two-tailed Student's unpaired *t*-test). AP (area postrema), CC (central canal), DMX (dorsal motor nucleus of the tenth cranial nerve, vagus), NTS (nucleus of the tractus solitarius). Scale bars indicate 100 μm. Source data are provided as a Source Data file.

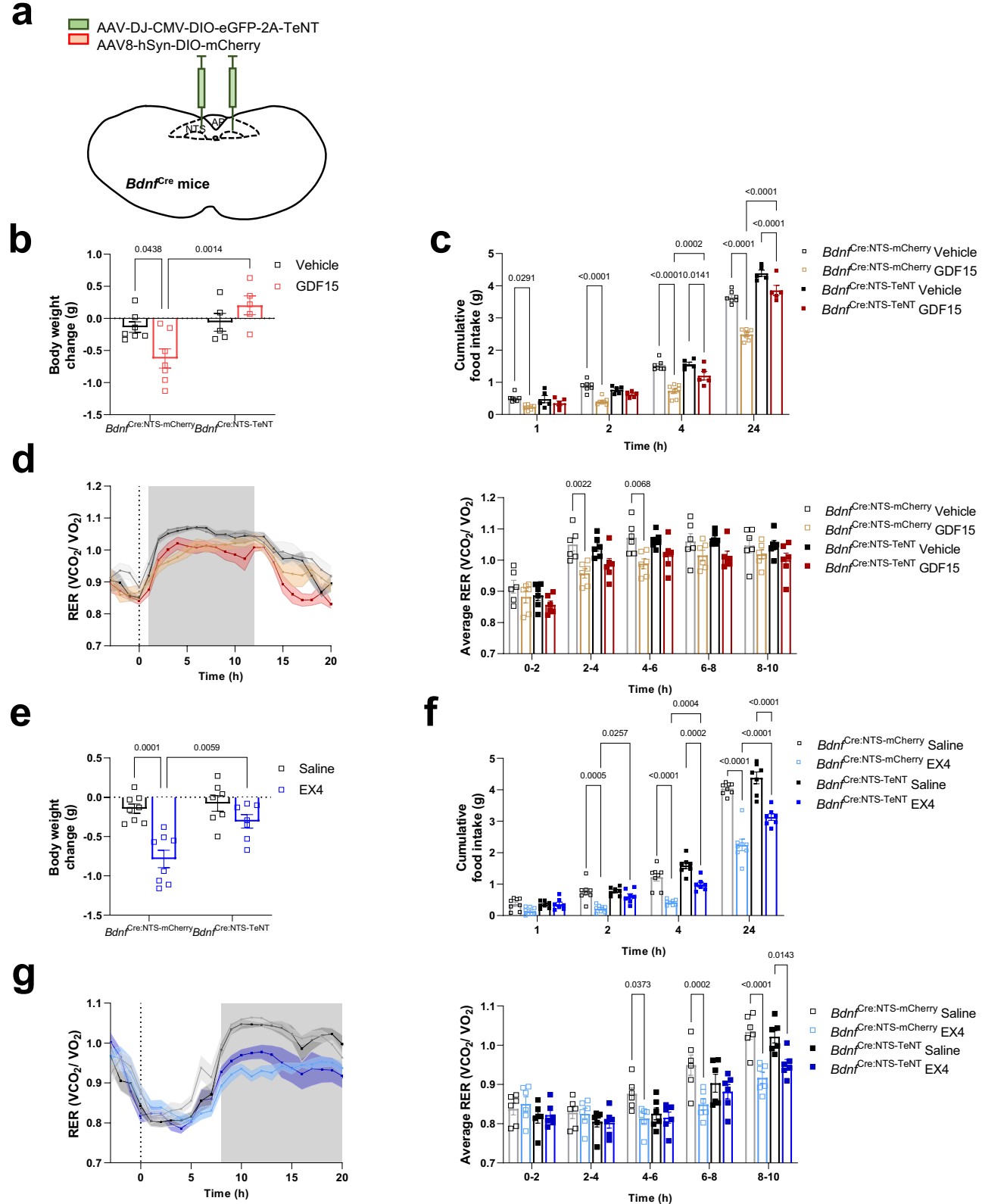

appear to be mediated partly by a direct projection from GFRAL neurons to CGRP neurons in the PBN[10,24]. However, prolonged GDF15 levels accentuate weight loss by increasing fatty acid oxidation through sympathetically mediated futile cycling in striated muscle rather than by maintaining anorexia[16]. Although to date there is limited support for the use of long-term GDF15 for body-weight reduction in obese humans[46], the mechanism of GDF15's action elucidated in preclinical

models could have important implications for other chronic weight management treatments. For example, although GLP1RA are believed to act primarily through reductions in food intake, there is evidence from preclinical models that the food intake effects of GLP1RA can decrease with chronic administration[42–45] and, therefore, other effects such as on energy expenditure or fatty acid oxidation may play an important role in maintaining weight loss without inducing anorexia or nausea.

**Fig. 5 | BDNF$^{mNTS}$ neurons are required for the metabolic effects of GDF15 or EX4. a** Schematic illustrating bilateral injection of AAV encoding the tetanus toxin light chain (green) or an AAV-mCherry control (red) into the mNTS of *Bdnf*$^{cre}$ mice. Based on the Paxinos and Franklin Mouse Brain Atlas[56]. **b** Body weight and (**c**) cumulative night-time food intake in *Bdnf*$^{Cre:NTS-TeNT}$ (*n* = 5) and *Bdnf*$^{Cre:NTS-mCherry}$ control (*n* = 7) mice following treatment with GDF15 (8 nmol kg$^{-1}$); (**b**) vehicle black squares and GDF15 red squares; *\*p* = 0.0438, *\*\*p* = 0.0014, (**c**) *\*p* = 0.0291, *\*p* = 0.0141, *\*\*\*p* = 0.0002,*\*\*\*\*p* < 0.0001; two-way ANOVA with Tukey's multiple comparisons test. **d** Indirect calorimetry demonstrated a significant reduction in RER following GDF15 in the *Bdnf*$^{Cre:NTS-mCherry}$ control mice, which did not occur in the *Bdnf*$^{Cre:NTS-TeNT}$ mice (8 nmol kg$^{-1}$; *n* = 6 per group, *\*\*p* = 0.0022, *\*\*p* = 0.0068; two-way ANOVA with Tukey's multiple comparisons test. **e** Body weight change and (**f**) cumulative night-time food intake in *Bdnf*$^{Cre:NTS-TeNT}$ (*n* = 7) and *Bdnf*$^{Cre:NTS-mCherry}$

control (*n* = 8) mice following treatment with EX4 (30 μg kg$^{-1}$); (**e**) vehicle black squares and EX4 blue squares;*\*\*p* = 0.0059, *\*\*\*p* = 0.0001, (**f**) *\*p* = 0.0257, *\*\*\*p* = 0.0005, *\*\*\*p* = 0.0002, *\*\*\*p* = 0.0002, *\*\*\*\*p* < 0.0001, two-way ANOVA with Sidak's multiple comparisons test (left), two-way ANOVA with Tukey's multiple comparisons test (right). **g** Indirect calorimetry assessment of RER in *Bdnf*$^{Cre:NTS-TeNT}$ and *Bdnf*$^{Cre:NTS-mCherry}$ mice following EX4 (*n* = 6 per group; *\*p* = 0.0373, *\*\*\*p* = 0.0002, *\*p* = 0.0143, *\*\*\*\*p* < 0.0001; two-way ANOVA with Tukey's multiple comparisons test. **c**, **d** *Bdnf*$^{Cre:NTS-mCherry}$ vehicle black open squares, GDF15 brown open squares; *Bdnf*$^{Cre:NTS-TeNT}$ vehicle black closed squares, GDF15 red closed squares. **f**, **g** *Bdnf*$^{Cre:NTS-mCherry}$ saline black open squares, EX4 light blue open squares; *Bdnf*$^{Cre:NTS-TeNT}$ vehicle black closed squares, EX4 dark blue closed squares. All data presented as mean ± SEM. Source data are provided as a Source Data file.

Here we show that GFRAL neurons, which contain GLP1R, project directly to CGRP$^{elPBN}$ neurons, but also locally to a distinct population of BDNF neurons in the mNTS. There are five *Bdnf*-containing neuronal clusters in the dorsal vagal complex, one of which corresponds with the GFRAL$^{AP/NTS}$ cells described here, and another that corresponds with PrRP/Calcr/Glp1r neurons in the NTS[27]. While we find EX4 (30 μg kg$^{-1}$) causes a robust activation of BDNF$^{mNTS}$ neurons, neither EX4 nor GDF15[10]activate PrRP$^{NTS}$ neurons. Likewise, ref. 27 found that EX4 (150 μg kg$^{-1}$) activated very few Calcr neurons in the NTS. The elPBN projection is part of a well-characterised pathway, involving further downstream structures such as the CeA and BSTov[21–23], and which is responsible for mediating the nausea-producing adverse effect of GLP1RA[19,25]. By comparison, the population of BDNF$^{mNTS}$ neurons activated by GDF15 or GLP1RA may instead relay information to the hypothalamic PVH. CRH neurons in the PVH, which lie at the head of the sympathetic and hypothalamo-adrenal axes, are activated by GDF15 or GLP1RA[10,29,30], while both corticosterone and adrenaline are released following activation of GFRAL neurons[31]. This is relevant as the fatty acid oxidative effect of GDF15 requires engagement of the sympathetic nervous system[12,13,16].

Disabling BDNF$^{mNTS}$ neurons with tetanus toxin resulted in body weight gain in male mice, but this did not reach significance in female mice. However, disabling the neurons attenuated EX4-induced weight loss in both males and females. Therefore, further in-depth studies will be required to determine if there are any true sex differences in physiology or if BDNF$^{mNTS}$ neurons have a role in normal, homeostatic regulation of body weight. Interestingly, it was possible to increase fatty acid oxidation by driving the activity of these neurons artificially, without a concomitant effect on energy expenditure. However, as the BDNF$^{mNTS}$ neurons were driven harder, energy expenditure and body temperature did decrease significantly. Thus, the mice were progressing towards a state of torpor. This is a natural survival strategy to protect energy stores and mice will enter torpor if food is unavailable and they are in an environment below thermoneutrality[47,48]. Brain circuits regulating torpor have been identified in the hypothalamus and preoptic area of the forebrain[49–52] and rostral brainstem[53] but ours and other recent data[31], suggest caudal brainstem neurons are also capable of driving this state.

Both the GFRAL→CGRP$^{elPBN}$ and GFRAL→BDNF$^{mNTS}$ pathways appear to be necessary for the full metabolic effects of GDF15 and GLP1RA. It will be interesting to look at the role of these two distinct pathways in longer-term treatment with clinically relevant GLP1RA and/or long-acting GFRAL agonists. We and others have demonstrated that blocking the GLP1RA-induced GFRAL→CGRP$^{elPBN}$ pathway with another incretin, glucose-dependent insulinotropic polypeptide, removes the aversive element of treatment without reversing the anorexia[19,25], which has important implications in the development of obesity drugs. Thus, the dual incretin receptor agonist, tirzepatide is effective at reducing weight in obese humans, apparently without the adverse nausea effects often reported with selective GLP1R agonists[54]. Why dual incretin

agonists should be more efficacious at reducing body weight, however, is unknown. Blocking the GFRAL→CGRP$^{elPBN}$ pathway may unmask the effectiveness of alternative pathways in reducing body weight. These alternatives may include the GFRAL→BDNF$^{mNTS}$ projection, described herein, or other pathways that respond to GLP1R agonism, either in the brainstem or elsewhere[45]. Finally, it is unclear whether the two GFRAL pathways we have described involve separate neurons or collaterals from the same neurons. A full understanding of the pathways mediating the various effects of anti-obesity drugs will hopefully lead to better treatments in the future.

## Methods
### Animals
Non-transgenic C57BL/6J mice were obtained from Charles River (Manston, Kent, UK) or Janvier (Laval, France). R26R-EYFP, ChR2-eYFP, *Bdnf*$^{Cre}$ and Ai95D (GCaMP6) mice were purchased from Jackson Laboratories (B6.129×1-Gt(ROSA)26Sor$^{tm1(EYFP)Cos}$/J); stock # 006148, B6;129S-Gt(ROSA)26Sor$^{tm32(CAG-COP4*H134R/EYFP)Hze}$/J; stock # 012569, B6.FVB-Bdnf$^{em1(cre)Zak}$/J; stock # 03018 and B6;129S-Gt(ROSA) 26Sor$^{tm95.1(95.1(CAG-GCaMP6f)Hze}$/J; stock # 024105 respectively; Jackson Laboratories, Maine, USA). *Calca*$^{Cre-GFP}$ mice were kindly donated by Dr Richard Palmiter[21] (B6.Cg-Calca$^{tm1.1(cre/EGFP)Rpa}$/J, Hughes Medical Institute, University of Washington, USA) and subsequently bred in-house. The *Prlh*$^{Cre}$ (previously described[55]; B6;129Sv/Pas-Prlh$^{tm1(cre)Smln}$) and *Gfral*$^{Cre}$ mouse lines were generated by GenOway (GenOway, Lyon, France). Briefly, for the generation of *Prlh*$^{Cre}$ mice, a *Plrh*$^{Cre}$ targeting vector containing an IRES-Cre transgene was constructed and inserted just downstream of the endogenous STOP codon in *Prlh* exon 2. For *Gfral*$^{Cre}$ generation, homology fragments encompassing the *Gfral* mouse gene were isolated from C57BL/6N mouse genomic DNA. A *Gfral*-Cre targeting vector was constructed containing an IRES-Cre transgene inserted just downstream of the STOP codon in *Gfral* exon 9. The targeting construct was transfected into embryonic stem cells, and correctly targeted clones were injected into blastocysts. High-percentage male chimeras (chimerism rate >50%) were mated with wild-type C57BL/6J mice to produce heterozygous offspring and F1 mice were identified by PCR. LoxSTOPlox-hM3Dq-T2A-mCherry (described as hM3Dq-mCherry) mice were generated using the TAR-GATT technology at Applied StemCell (Milpitas, CA, USA; described previously[34]). Briefly, an integration cocktail consisting of a plasmid donor DNA vector containing frt-STOP-frt-loxP-STOP-loxP-hM3Dq-T2A-mCherry flanked by two attB sites and in vitro transcribed φC31 integrase mRNA was microinjected into the pronuclei of heterozygous embryos harvested from B6 mice previously engineered with three tandem attP sequences into mouse H11 locus. Mice were subsequently crossed with a pan FLPe-expressing mouse line (B6.129S4-Gt(ROSA) 26Sor$^{tm1(FLP1)Dym}$/RainJ, strain # 009086) to remove the frt-STOP-frt cassette and generate the parent Cre-inducible hM3Dq-mCherry line, which is bred in house. Mice were group housed unless otherwise stated in a 12:12 h light: dark cycle, at room temperature (22 °C; 50–55%

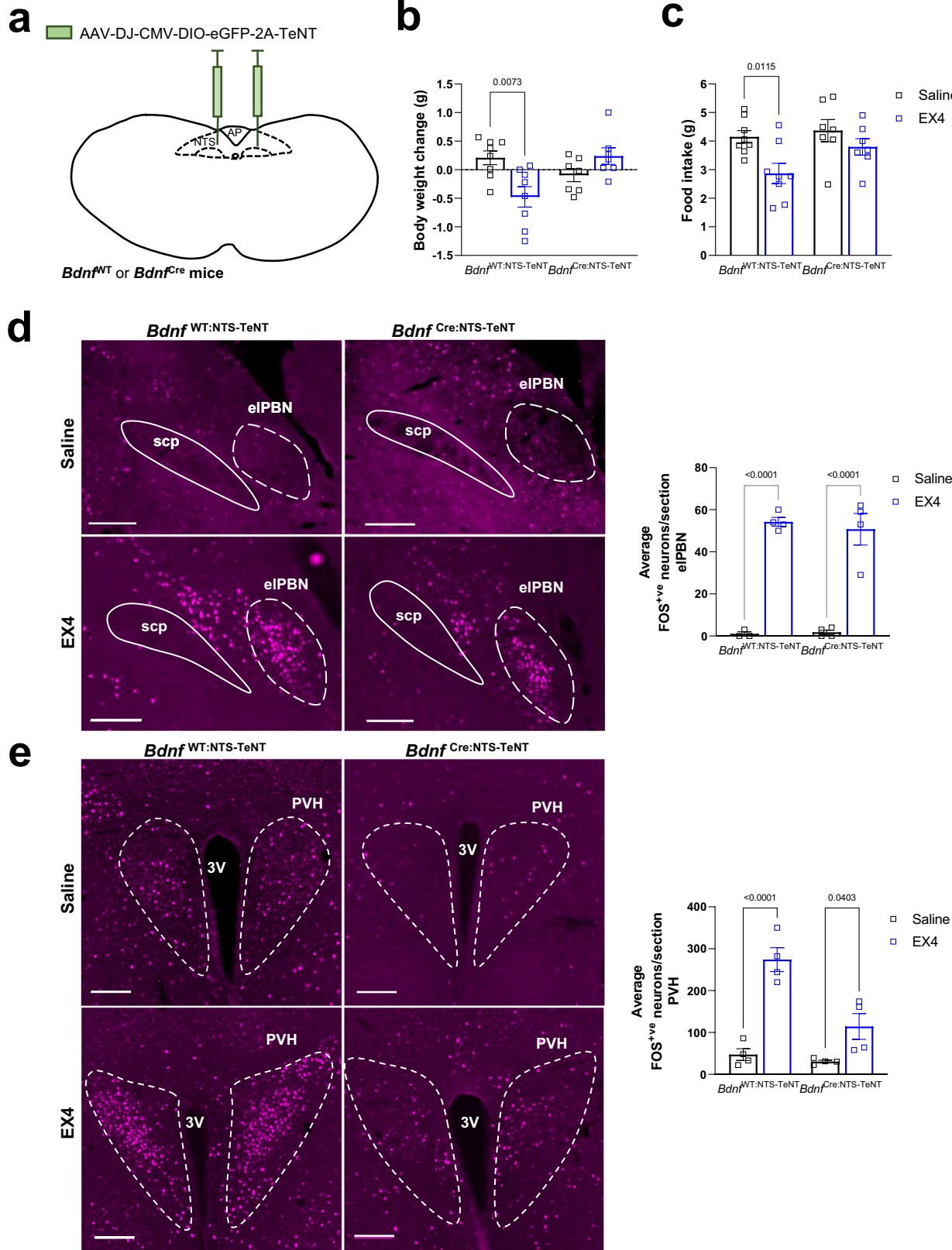

humidity). Mice were provided with *ad libitum* access to water and standard rodent chow unless otherwise stated. All animal studies were performed in accordance with the UK Animals and Scientific Procedure Act, 1986, and approved by a local ethics committee, and in compliance with Eli Lilly and Company's Institutional Animal Care and Use Committee.

**Drugs and viral vectors**

GDF15 (Cat.# 9279-GD; R&D Systems) was initially dissolved in 15 mM HCl, neutralised with 7.5 mM NaOH, and diluted in saline and vehicle was made up in the same way. GDF15 was injected subcutaneously at either 4 or 8 nmol kg$^{-1}$ in a volume of 4 ml kg$^{-1}$. Semaglutide was supplied by Eli Lilly (Indianapolis, USA), and was dissolved directly in saline

**Fig. 6 | Disabling BDNF^mNTS neurons prevents EX4-mediated activation of the PVH, but not other downstream targets. a** Schematic illustrating bilateral injection of AAV encoding the tetanus toxin light chain into the mNTS of *Bdnf*^cre mice and *Bdnf*^WT control mice. Based on the Paxinos and Franklin Mouse Brain Atlas[56]. **b** Body weight change and (**c**) cumulative food intake in *Bdnf*^Cre:NTS-TeNT (*n* = 7) and *Bdnf*^WT control (*n* = 8) mice 24 h after treatment with EX4 (30 μg kg⁻¹; two-way ANOVA with Tukey's multiple comparisons test; **$p$ = 0.0073, *$p$ = 0.0115 respectively). Fluorescent staining of EX4-induced FOS (magenta) in (**d**) the elPBN and (**e**)

PVH of *Bdnf*^Cre:NTS-TeNT and *Bdnf*^WT control mice. The average number of FOS+ve neurons/section is presented on the right (*n* = 4 per group; two-way ANOVA with Sidak's multiple comparisons test; *$p$ = 0.0403, ****$p$ < 0.0001). 3 V (third ventricle), elPBN (exterolateral parabrachial nucleus), PVH (paraventricular nucleus of the hypothalamus), scp (superior cerebellar peduncle). All data presented as mean ± SEM. Saline black squares and EX4 as blue squares throughout Scale bars indicate 200 μm and 500 μm in the elPBN and PVH, respectively. Source data are provided as a Source Data file.

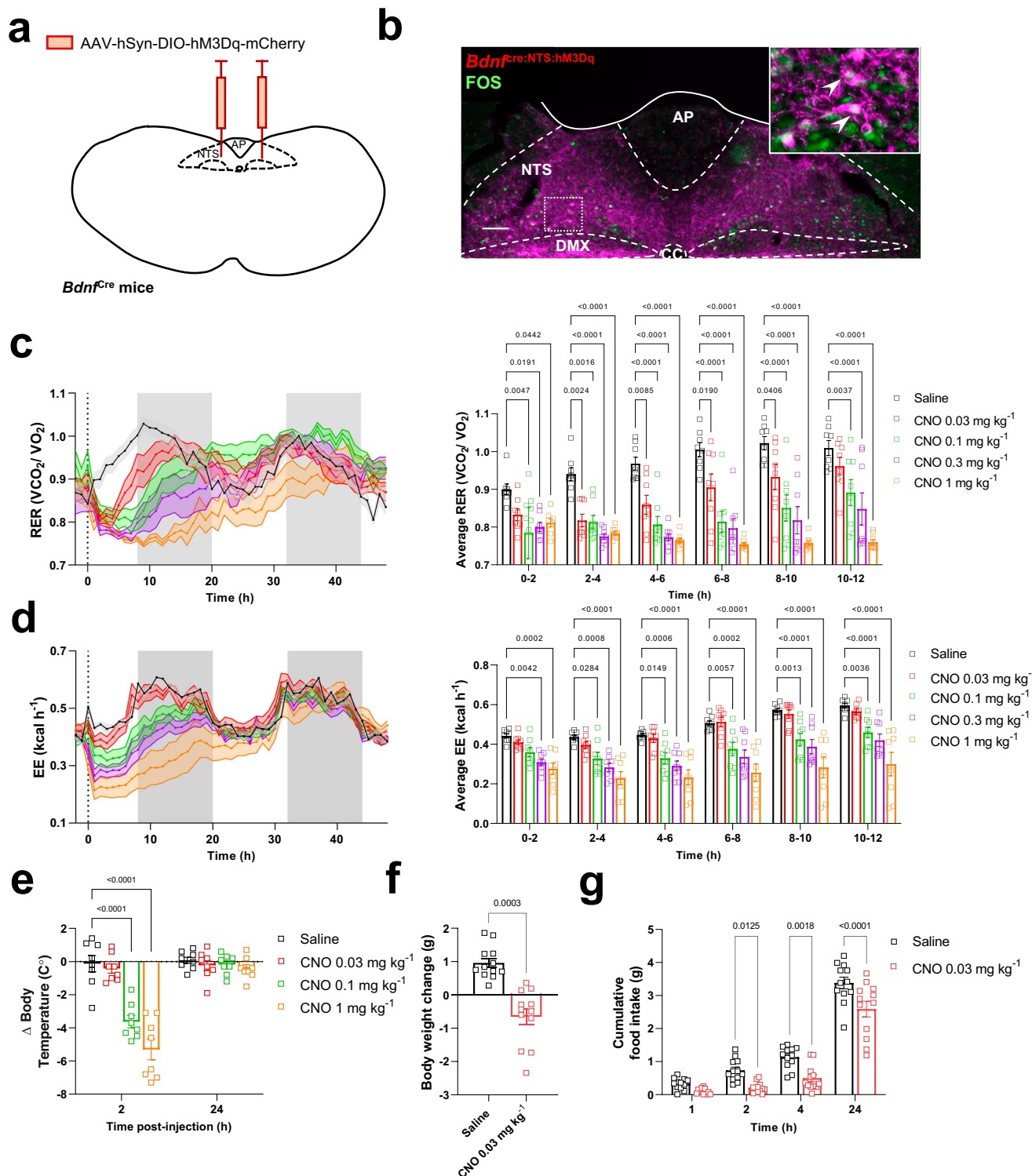

**Fig. 7 | Acute activation of BDNF^mNTS neurons is sufficient to reduce food intake and RER. a** Schematic illustrating bilateral injection of AAV-hSyn-DIO-hM3Dq-mCherry into the mNTS of *Bdnf*^Cre mice. Based on the Paxinos and Franklin Mouse Brain Atlas[56]. **b** Representative immunofluorescence image of AAV-hSyn-DIO-hM3Dq-mCherry (magenta) into the mNTS of *Bdnf*^Cre mice with FOS (green) induction following CNO administration (vehicle $n = 6$, CNO $n = 6$). Scale bar indicates 100 μm. **c** Indirect calorimetry assessment of RER (*$p = 0.0442$, *$p = 0.0191$, ***$p = 0.0047$, ***$p = 0.0016$, ***$p = 0.0024$, ***$p = 0.0085$, *$p = 0.0190$, *$p = 0.0406$, ***$p = 0.0037$, ****$p < 0.0001$) and (**d**) energy expenditure (***$p = 0.0042$, ***$p = 0.0002$, *$p = 0.0284$, ***$p = 0.0008$, *$p = 0.0149$, ***$p = 0.0006$, **$p = 0.0057$, ***$p = 0.0002$, ***$p = 0.0013$, ***$p = 0.0036$, ****$p < 0.0001$) in *Bdnf*^Cre:NTS-hM3Dq mice following increasing doses of CNO (0.03 mg kg⁻¹, 0.1 mg kg⁻¹, 0.3 mg kg⁻¹ and 1 mg kg⁻¹; $n = 8$ per group; two-way ANOVA with Dunnett's multiple comparisons).

**e** Body temperature change in *Bdnf*^Cre:NTS-hM3Dq mice 2 h and 24 h after CNO (0.03 mg kg⁻¹, 0.3 mg kg⁻¹, 1 mg kg⁻¹) using a thermal imaging camera ($n = 8$ per group; ****$p < 0.0001$; two-way ANOVA with Sidak's multiple comparisons). **f** Body weight change in *Bdnf*^Cre:NTS-hM3Dq mice 24 h after treatment with saline or CNO (0.03 mg kg⁻¹; $n = 12$ per group; two-tailed Student's paired *t*-test; ***$p = 0.0003$). **g** Cumulative night-time food intake in *Bdnf*^Cre:NTS-hM3Dq mice following treatment with saline or CNO ($n = 12$ per group, two-way ANOVA with Sidak's multiple comparisons test; *$p = 0.0125$, **$p = 0.0018$, ****$p < 0.0001$). AP (area postrema) CC (central canal), DMX (dorsal motor nucleus of the tenth cranial nerve, vagus), NTS (nucleus of the tractus solitarius). All data presented as mean ± SEM. Saline black squares, CNO 0.03 mg kg⁻¹ red squares, CNO 0.1 mg kg⁻¹ green squares, CNO 0.3 mg kg⁻¹ purple squares and CNO 1 mg kg⁻¹ orange squares throughout. Source data are provided as a Source Data file.

(10 nmol kg⁻¹, 4 ml kg⁻¹ intraperitoneally (IP)). For in vivo studies Exendin-4 (EX4) was supplied by Eli Lilly and was dissolved directly in saline (30 μg kg⁻¹, 10 ml kg⁻¹ intraperitoneally (IP)). For in vitro calcium imaging experiments stock solutions of EX4 (Cat.# 1933; Tocris Bioscience, Bristol, UK) were prepared in double-distilled $H_2O$ before addition into the recording solution. For chemogenetic stimulation experiments, clozapine *N*-oxide (CNO; Cat.# 4936; Tocris Bioscience) was dissolved directly in saline. For Gfral^Cre:hM3Dq-mCherry stimulation in male and female mice (10–16 weeks old) CNO was injected IP (3 mg kg⁻¹, 4 ml kg⁻¹). For other chemogenetic stimulation experiments CNO was injected IP at 0.03–1 mg kg⁻¹, 4 ml kg⁻¹.

For all NTS injections, mice were injected using a modified stereotaxic injection procedure as previously described[35]. Briefly, mice were anaesthetised with a mixture of ketamine and xylazine dissolved in saline (80 and 10 mg kg⁻¹, respectively; 10 ml kg⁻¹ IP). Mice were placed in a stereotaxic frame, an incision was made at the level of the *cisterna magna*, and neck muscles were carefully retracted. Following an incision in the *dura*, the *obex* served as a reference point for injections using a glass micropipette attached to a Nanoject II Auto Nano-liter Injector (Drummond Scientific Company, PA, USA). NTS coordinates were -0.2 mm A/P, ±0.2 mm M/L, −0.2 mm D/V from *obex*. All animals were administered analgesia (5 mg kg⁻¹ Carprofen) subcutaneously for 2 days post operatively.

For neuronal silencing studies, AAV-DJ-CMV-DIO-eGFP-2A-TeNT (Cat.# AAV-71; Stanford Vector and Viral Core; Stanford University, USA), was injected bilaterally into the NTS of 10-week old male and female *Bdnf*^Cre or wildtype littermates as a control, at a total volume of 207 nl per side. In other experiments, AAV8-hsyn-DIO-mCherry (207 nl per side; Cat.# 50459-AAV8; Dr Bryan Roth, Addgene, MA, USA) was used as a control virus in Cre-expressing mice. For neuronal activation studies in 18-week old male *Bdnf*^Cre mice, AAV9-hsyn-DIO-hM3Dq-mCherry (Cat.# 44361-AAV9; Dr Bryan Roth, Addgene) was injected bilaterally into the NTS in a total volume of 60 nl per side. A titre of $1 \times 10^{12}$ was used for all viral injections. Body weight (g) was measured weekly post-surgery and food intake was measured daily over a period of 5 days at 7 weeks post surgery. Feed efficiency ratio (FER) was calculated using the following equation: FER = (body weight gain (g)/food intake (g))*100.

For retrograde tracing studies, FluoroGold (hydroxystilbamidine, 4% w/v solution in water; Invitrogen, ThermoFisher, MA, USA), was injected unilaterally, at a total volume of 19 nl, into the NTS of 12-week-old C57BL/6J male mice ($n = 8$). FluoroGold was also injected into the lateral PBN and CeA of 14-week old male C57BL/6J (total volume 18 nl, $n = 5$) and 11–12 week old male and female mice (total volume 18 nl, $n = 5$), respectively, using standard stereotaxic procedures under iso-flurane anaesthesia (2–3% oxygen). Co-ordinates were determined from the Mouse Brain Atlas[56]. Lateral PBN coordinates from *bregma*: −4.9 mm A/P, −1.4 mm M/L, −3.8 mm DV and CeA coordinates from *bregma*: −1.4 mm AP, −2.7 mm M/L, −5.1 mm D/V. After allowing two weeks for recovery and axonal transport, mice were transcardially perfused (see below).

## Optogenetics
Male *Gfral*^Cre:ChR2-eYFP mice (8–16 weeks old) were anaesthetized with isoflurane (2–3% in oxygen) and placed in a stereotaxic frame. The skull was exposed, and a hole was drilled at the site of implantation. Mice were unilaterally implanted with an optic fibre 200 μm, 0.39NA (Thorlabs, NJ, USA) above the PBN −5.0 mm A/P, −1.5 mm M/L, −3.5 mm D/V from *bregma*, and the fibre was secured using dental cement. Mice were allowed to recover from surgery for at least two weeks. Mice were then habituated to handling and tethering and acclimated to open cages for at least one hour per day for a minimum of four days before the start of the experiment. Mice had their food removed at the onset of the dark phase on the night before the experiment. At 10 am the following morning, mice were tethered to fibre optic cables (200 μm; Plexcon Inc., Texas, USA) using zirconia sleeves (Doric Lenses, Franquet, Quebec, Canada) to firmly attach to the fibre implant, for 10 min prior to stimulation. Photostimulation was programmed using a pulse generator software (Radiant V2) that controlled a blue light LED (465 nm; Plexon PlexBright; Plexon) via a Plexon LED Controller. Photostimulation was set as light pulse trains with 10 ms pulse width at 10 Hz. Mice were optogenetically stimulated for 60 min and food intake measured at 15 min, 30 min, and 60 min following the start of the stimulation protocol. The light was adjusted such that the power exiting the fibre optic cable was at least 9 mW mm⁻², measured using a digital optical power meter (PM100D, Thorlabs) and an online light transmission calculator for brain tissue (http://web.stanford.edu/group/dlab/cgi-bin/graph/chart.php).

## Metabolic phenotyping
In all metabolic phenotyping studies, mice were singly housed for at least one week before being put into indirect calorimetric cages of either of two systems. In the Comprehensive Laboratory Animal Monitoring System (CLAMS; Columbus Instruments, Columbus, OH, USA), oxygen consumption (VO₂) and carbon dioxide production (VCO₂) were measured every 8 min using Oxymax® software (Columbus Instruments). In the PhenoMaster (TSE Systems, Bad Homburg, Germany), VO₂ and VCO₂ were measured, along with food weight (g), every 2 min using LabMaster software (TSE Systems). Calorimetric cages were not equipped with running wheels, and environmental enrichment was limited to bedding material. During this time all mice had *ad libitum* access to food and water unless otherwise stated. Mice were allowed to acclimate to the indirect calorimetric cages for a minimum of three days, and baseline recordings were made for a minimum of three days before the start of treatment.

In all experiments, VO₂ (ml h⁻¹) and VCO₂ (ml h⁻¹) were used to calculate the RER (RER = VCO₂/VO₂) and EE (EE = [3.941 * VO₂ + 1.106 * VCO₂]/1000 in kcal h⁻¹). Whole-body fat utilization was calculated using the following equation: $1.67 \times (VO_2 - VCO_2)$[57]. Data were averaged every 60 min for continuous plots and averaged every two hours for comparisons.

## Acute food intake studies

For all feeding studies mice were acclimated to single housing for at least one week prior to the start of the experiment. For night-time feeding studies, on the day of the experiment, food was removed from the animals for 3 h before the onset of the dark phase. At lights out, mice were administered drugs and food was returned. Food intake was recorded at 0, 1, 2, 4 and 24 h after treatment. For fast-refeeding studies, mice had their food removed at the onset of the dark phase on the night before the experiment. At 10 am the following morning, mice were administered drugs and food was returned. Food intake was measured at 0, 1, 2, 4 and 24 h after injection of drugs. In crossover experiments, mice were allowed to recover for a minimum one week before the alternative treatment.

## Conditioned taste avoidance test

For CTA, mice were singly house and acclimated to water bottles in the home cage for a minimum of two weeks prior to the start of the experiment. For at least four consecutive days mice were habituated to cages that permitted *ad libitum* access to water from two bottles, side-by-side for an hour per day. On day 1 of the study, animals were water-deprived overnight for 16 h. The following morning (day 2), water-deprived animals were provided with a bottle of novel 15% sucrose solution (dissolved in drinking water) for 30 min. At the end of the 30 min sucrose exposure, each animal received an IP injection of saline or CNO (3 mg kg$^{-1}$) or an IP injection of vehicle or semaglutide (10 nmol kg$^{-1}$). The mice were then returned to a normal home environment and had unlimited access to water for one night. On day 3, mice were again water-deprived overnight. On day 4, water-deprived animals were provided with one bottle of 15% sucrose and one bottle of water for a period of 24 h. Volumes of sucrose and water intake were measured at 2 h and used to calculate sucrose preference (sucrose intake/total fluid intake * 100). Food was available *ad libitum* throughout the study. Sucrose and water bottle position (left or right) was randomised within treatment groups.

## Gastric emptying

To assess the rate of gastric emptying the acetaminophen absorption test was used. *Gfral*$^{Cre:hM3Dq-mCherry}$ mice were habituated to handling and restraint tubes for at least 2 weeks prior to the experiment. Mice were fasted for 3 h and administered IP saline or CNO (3 mg kg$^{-1}$) two hours prior to oral administration of a 1.5% glucose solution that contained acetaminophen (100 mg kg$^{-1}$; Sigma-Aldrich, UK). Mice were warmed in a hot box (36 °C) prior to tail vein bleeds. 50 µl blood was collected into heparin-coated tubes at 0, 15, 30, and 60 min after acetaminophen gavage. Blood samples were stored on ice for a minimum of 30 min before plasma separation by centrifugation at 4 °C, and plasma was subsequently stored at −80 °C. An enzymatic-spectrophotometric assay was used with a modified protocol for low volume samples (paracetamol/acetaminophen assay kit K8002; Cambridge Life Sciences, UK).

## Thermal imaging

For thermal imaging studies, individual mice were placed into an open cage containing only woodchip material and 10 s videos were recorded from above using a FLIR A655sc high-resolution LWIR thermal camera and FLIR Research IR software (Teledyne FLIR LLC, Oregano, USA). Videos were recorded before treatment and 2 and 24 h after treatment. Images from the recordings were analysed using Research IR software, by drawing regions of interest (ROI) and the maximal value (°C) from the ROI was used to calculate the difference in body temperature.

## Tissue preparation and histology

Before brains were collected, most animals used for FOS studies were fasted at 9 am and administered drugs 2 h prior to culling. *Prlh*$^{Cre::EYFP}$ male mice (9–13 weeks old) were fasted overnight and saline or EX4 (IP, 30 µg kg$^{-1}$, $n = 6$ per group) administered 90 min prior to culling.

For all immunohistochemistry, mice were deeply anaesthetised with 4% isoflurane in oxygen and transcardially perfused with heparinsed saline (20,000 U per litre in 0.9% NaCl) followed by 4% paraformaldehyde in 0.1 M phosphate buffer. Brains were dissected and post-fixed overnight at 4 °C and then cryoprotected in 30% sucrose. Brains were cut into 30 µm-thick coronal sections using a freezing sledge microtome (Bright 8000, Cambridge, UK) and either processed immediately or stored in cryoprotectant solution at −20 °C.

Immunohistochemistry was performed on free-floating sections at room temperature. Brain sections were washed in 0.2% Triton X-100 in 0.1 M phosphate buffer and blocked in 5% normal serum for 1 h, before being incubated in primary antibody (sheep polyclonal anti-GFRAL (dilution 1:200, Cat. # PA5-47769, Thermo Fisher Scientific, MA, USA), rabbit polyclonal anti-cFos (dilution 1:2500 or 1:4000 Cat. #190289, Abcam), chicken polyclonal anti-GFP (dilution 1:2000, Cat. #13970, Abcam, Cambridge, UK), goat anti-mCherry (dilution 1:5000, #AB0040-200, Sicgen, Cantanhede, Portugal)); made up in to 1% normal serum overnight at 4 °C. The next day, sections were washed again and then incubated in secondary antibody at room temperature (donkey polyclonal anti-sheep Alexa Fluor 594 (dilution 1:1000, Cat.# A11016 Molecular Probes, OR, USA), donkey polyclonal anti-chicken Alexa Fluor 488 (dilution 1:1000, Cat.# 703-545-155, Jackson ImmunoResearch, PA, USA), donkey polyclonal anti-rabbit Alexa Fluor 488 (dilution 1:1000, Cat.# 711-545-152, Jackson ImmunoResearch), donkey polyclonal anti-rabbit Alexa Fluor 594 (dilution 1:1000, Cat.# 711-585-152, Jackson ImmunoResearch) for 2 h, or biotinylated donkey polyclonal anti-rabbit (dilution 1:1000, Cat.# 711-065-152, Jackson ImmunoResearch) overnight at 4°C followed by streptavidin Alexa Fluor 594 (dilution 1:1000, Cat.# 016-580-084, Jackson ImmunoResearch) for 2 h). Finally, sections were washed in water, mounted onto glass slides, air-dried overnight and coverslipped with ProLong Gold (Thermo Fisher Scientific). Images were acquired on a 3D-Histech Panoramic-250 microscope slide scanner using a 20×/0.80 Plan Apochromat objective (Zeiss, Baden-Württemberg, Germany) and FITC and Texas Red filter sets. Snapshots of the slide scans were taken using the SlideViewer software (3D-Histech, Budapest, Hungary). Further images were collected on a Zeiss Axioimager.M2 upright microscope using a 10× Plan Apochromat objective and captured using a Coolsnap HQ2 camera (Teledyne Photometrics, CA, USA) through Micromanager software v1.4.23. Specific band pass filter sets for DAPI, FITC and Texas Red were used to prevent bleed through from one channel to the next. Images were processed and analysed using Fiji ImageJ (http://imagej.net/Fiji/Downloads). For FOS quantification sections were counted manually, and averaged either per section, or per side per section, as specified.

For RNAscope in situ hybridization histology, 12-week-old male mice were fasted at the time of injection. 1 h later mice were culled and dissected brains were frozen immediately on dry ice. Multiplex RNAscope in situ hybridization histology for mouse *c-fos* (Cat.# 316928-C3, Advanced Cell Diagnostics, CA, USA), mouse *Gfral* (Cat.# 417028-C2, Advanced Cell Diagnostics) and mouse *Bdnf* (Cat.# 424828-C1, Advanced Cell Diagnostics) was performed on 8 µm cryo-sections covering the AP/NTS according to manufacturer's instructions (Advanced Cell Diagnostics, CA, USA) at Gubra (Horshom, Denmark). After hybridization, sections were counterstained with DAPI and coverslipped using a fluorescence mounting medium. Slides were scanned under a 20× objective in an Olympus VS-120 slide scanner (Olympus Corporation, Tokyo, Japan) with appropriate fluorescent filters. Quantification of the number of *Gfral*+ve or *Bdnf*+ve cells co-expressing *c-fos* was performed blinded at The University of Manchester.

## Calcium imaging

Male and female *Gfral*$^{Cre:Ai95D}$ and *Bdnf*$^{Cre:Ai95D}$ mice (10–26 weeks old) were culled through cervical dislocation and the brain was removed into an ice-cold incubation solution composed of (mM): 95 NaCl, 1.8 KCl, 1.2 KH$_2$PO$_4$, 7 MgSO$_4$, 26 NaHCO$_3$, 15 glucose, 50 sucrose, 0.5

$CaCl_2$ and bubbled with 95% $O_2$/5% $CO_2$. Coronal brainstem slices (230 μm thick) were prepared using a vibratome (7000smz; Campden Instruments, Loughborough, UK). Slices were maintained in room temperature incubation solution for up to 1 h before transfer to a room temperature recording solution composed of (mM): 127 NaCl, 1.8 KCl, 1.2 $KH_2PO_4$, 1.3 $MgSO_4$, 26 $NaHCO_3$, 5 glucose, 0.5 $CaCl_2$ and bubbled with 95% $O_2$/5% $CO_2$, for 1 h prior to recording.

Brain slices were transferred to a submerged slice chamber and perfused with oxygenated room temperature recording solution at a flow rate of 2 ml min$^{-1}$. Slices were visualised using a fluorescence microscope (Axioskop 2, Zeiss), equipped with a 40×/0.8 immersion objective. Recordings were made using a Prime BSI Express sCMOS camera (Teledyne Photometrics). GCaMP6f in $Gfral^{AP}$, $Bdnf^{AP}$ and $Bdnf^{NTS}$ neurons was excited at 470 nm using a pE-400 light source (CoolLED Ltd) applied for 50 ms at 1 Hz. Baseline GCaMP6f fluorescence was recorded for 2 min before addition of EX4 (100 nM) to the recording solution for 2 min. KCl (50 mM) was added to the solution at the end of the experiment to induce a maximal response.

Fluorescence intensities in GCaMP6+ve cell bodies were quantified using Fiji ImageJ. All cells that responded to KCl were analysed. The fluorescence values of each cell over the entire recording were standardised using the standard Z-score formula:

$$Z = (X - \mu)/\sigma$$

where $X$ represents the raw fluorescence intensity for each time point, $\mu$ is the mean fluorescence intensity of the entire recording for each cell, and $\sigma$ is the standard deviation of the fluorescence values for that cell. The Z-score values were then adjusted by subtracting the mean Z-score during the baseline period prior to EX4 application, ensuring that baseline activity was centred around zero:

$$Z\,\text{final} = Z - \bar{Z}\text{baseline}$$

Peak calcium responses were taken from the 1 min following EX4 application and averaged for each slice. For statistical analysis, n was defined as the number of independent slices for each condition (not the number of cells).

## Statistics

Details of statistical analyses are provided in the corresponding figure legends and were carried out using Prism statistical package (Graph-Pad Software Inc, San Diego, USA). Data are displayed as mean ± SEM and are compared using parametric statistics, with $P$ values less than 0.05 considered as statistically different and denoted as follows: *$p < 0.05$, **$p < 0.01$, ***$p < 0.001$ and ****$p < 0.0001$. Two experimental groups were analysed using the Student's $t$-test and multiple groups were analysed using ANOVA with appropriate *post hoc* tests.

## Reporting summary

Further information on research design is available in the Nature Portfolio Reporting Summary linked to this article.

## Data availability

All data generated in this study are available in the article, supplementary information and the Source Data file. Source data are provided with this paper.

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

## Acknowledgements

The work was funded by the Biotechology and Biological Sciences Research Council Industrial Partnership Award BB/S008098/1 (SML) with Eli Lilly, BB/Z516405/1 (SML), and a Medical Research Council grant MR/T032669/1 (SML and GD).

## Author contributions

Conceptualization: S.M.L., P.J.E., T.C. Methodology: C.H.F., M.B.B., G.D. Investigation: C.H.F., V.C., A.A.W., R.S., S.G., C.H. Funding acquisition: S.M.L., P.J.E., T.C. Project management: S.M.L., P.J.E. Writing – original draft: S.M.L. Writing – review & editing: S.M.L., G.D., P.J.E.

## Competing interests

The authors declare the following competing interests: PJE and TC are paid employees of Eli Lilly. The other authors declare no competing interests.
