## [Transparent Peer Review file · Nature Communications]

Brainstem BDNF neurons are downstream of GFRAL/GLP1R signalling

Corresponding Author: Professor Simon Luckman

This manuscript has been previously reviewed at another journal. This document only contains reviewer comments, rebuttal and decision letters for versions considered at Nature Communications.

Version 0:

Reviewer comments:

Reviewer #1

(Remarks to the Author)

This is a transferred manuscript that was previously reviewed and has been modified by the authors in response to the raised critiques. My enthusiasm for this manuscript remains high and the authors have adequately addressed the critiques raised by the original review.

Reviewer #2

(Remarks to the Author)

The revised manuscript is improved over the original and most concerns have been effectively addressed. One remaining concern relates to the heterogeneous nature of BDNF-expressing neurons in the hindbrain, some of which have been characterized previously while others have not. As the authors point out in their response to reviewers, identifying the specific neuronal subset (or subsets) involved will require additional study. But the authors' response to this concern is unsatisfying in the sense that the data presented do not exclude the possibility that BDNF/PrLh/Calcr neurons, for example, play a role. The authors are quick to dismiss this possibility without really addressing it.

This concern does not negate the significance of the overall findings, but it leads to interpretive difficulties that should be addressed. Highlighting this point is the statement in the Abstract: "Here we describe a separate pathway downstream of GFRAL/GLP1R neurons that involves a distinct population of brain-derived neurotrophic factor cells in the medial nucleus of the tractus solitarius." The problem of course is that the authors have not identified a distinct population of BDNF neurons, so the statement is inaccurate. The authors should acknowledge throughout the manuscript the existence of multiple distinct BDNF neuron subsets in this brain area, and that additional study is needed to identify the specific population involved.

Prof Simon M. Luckman
Faculty of Biology, Medicine and
Health
A.V. Hill Building
The University of Manchester
Oxford Road
Manchester M13 9PT

Tel +44(0)161 275 5381
simon.luckman@manchester.ac.uk

4th October 2024.

NCOMMS-24-43518-T

Please find the response to reviewer's comments below:

Reviewer #1

The reviewer notes that we have addressed the critiques from the original review and their enthusiasm for the manuscript remains high.

Reviewer #2

We accept the concerns of the reviewer that the population of BDNF neurons in the NTS that are activated by GDF15 or Exendin-4 (EX4) could be a previously described population that contain PrRP/Calcr. This is despite our own paper showing that GDF15 does not activate PrRP neurons (Ref 10), and published results elsewhere (Ref 27) demonstrating that a very high dose of EX4 activates very few Calcr neurons. However, as the Reviewer maintains concern, we have carried out a separate experiment (new Suppl Figure 4) to demonstrate that EX4 does not activate PrRP neurons in the NTS. The text has been altered to include the new data, and the Suppl Figure numbers have been modified.

We have responded to the points raised by the editorial office and now have uploaded a manuscript with tracked changes. We have also uploaded a clean document along with the Figure files, author checklist, etc.

Yours sincerely,

Professor Simon Luckman
Brackenbury Chair of Physiology.